# Learning non-stationary Langevin dynamics from stochastic observations of latent trajectories

Mikhail Genkin [ID] [1], Owen Hughes[2] & Tatiana A. Engel [ID] [1✉]

Many complex systems operating far from the equilibrium exhibit stochastic dynamics that can be described by a Langevin equation. Inferring Langevin equations from data can reveal how transient dynamics of such systems give rise to their function. However, dynamics are often inaccessible directly and can be only gleaned through a stochastic observation process, which makes the inference challenging. Here we present a non-parametric framework for inferring the Langevin equation, which explicitly models the stochastic observation process and non-stationary latent dynamics. The framework accounts for the non-equilibrium initial and final states of the observed system and for the possibility that the system's dynamics define the duration of observations. Omitting any of these non-stationary components results in incorrect inference, in which erroneous features arise in the dynamics due to non-stationary data distribution. We illustrate the framework using models of neural dynamics underlying decision making in the brain.

---

[1] Cold Spring Harbor Laboratory, Cold Spring Harbor, NY, USA. [2] University of Michigan, Ann Arbor, MI, USA. ✉email: engel@cshl.edu

Many complex systems generate coherent macroscopic behavior that can be expressed as simple laws. Such systems are commonly described by Langevin dynamics, in which deterministic forces define persistent collective trends and noise captures fast microscopic interactions[1]. Langevin equations are used to model stochastic evolution of complex systems such as neural networks[2–5], motile cells[6], swarming animals[7], carbon nanotubes[8], financial markets[9], or climate dynamics[10]. While such systems can be readily observed in experiments or microscopic simulations, the analytical form of the Langevin equation usually cannot be easily derived from microscopic models or physical principles. The inference of Langevin equations from data is therefore crucial to enable efficient analysis, prediction, and optimization of complex systems.

Numerous methods were proposed for inferring Langevin dynamics from stochastic trajectories[11], e.g., by estimating moments of local trajectory increments[1,12–17]. However, in many complex systems, the trajectories cannot be observed directly, but are only gleaned from a stochastic observation process that depends on the latent Langevin dynamics[18]. For example, spikes recorded from neurons in the brain form stochastic point processes with statistics controlled by the collective dynamics of the surrounding network[2,19,20]. Similarly, the dynamics of a protein are observed through photons emitted by fluorescent dyes tagging the protein in single-molecule microscopy experiments[21–23]. The Poisson noise inherent in spike or photon observations makes the inference of the underlying Langevin dynamics challenging.

This challenge can be addressed by modeling data as a doubly stochastic processes, in which latent stochastic dynamics drive another stochastic process modeling the observations[24]. The inference with latent dynamical models is data efficient as it integrates information along the entire latent trajectory, but it may be sensitive to the data distribution. Previous work only considered the inference of latent Langevin dynamics for equilibrium systems with the steady-state data distribution[2,23]. Whether these methods extend to non-equilibrium systems has not been tested. Yet, all living systems and physical systems that perform computations operate far from equilibrium, where transient dynamics play a key role. The inference of non-stationary Langevin dynamics from stochastic observations remains an important open problem.

Here we present an inference framework for latent Langevin dynamics which accounts for non-equilibrium statistics of latent trajectories. We show that modeling non-stationary components is critical for accurate inference, and their omission leads to biases in the estimated Langevin forces. As a working example, we model non-stationary dynamics of neural spiking activity during perceptual decision making, a process of transforming a sensory stimulus into a categorical choice[25]. The inference of the underlying dynamics from spikes is notoriously hard[26], and analyses with simple parametric models result in controversial conclusions[27–30]. Our framework accurately infers the Langevin dynamics from spike data generated by competing models of decision making proposed previously. Our framework can be extended to different stochastic observation processes and is broadly applicable for the inference of the Langevin dynamics in non-stationary complex systems.

## Results

**Inference framework**. We consider the inference of Langevin dynamics

$$\frac{\mathrm{d}x}{\mathrm{d}t} = DF(x) + \sqrt{2D}\xi(t), \qquad (1)$$

where $F(x)$ is the deterministic force, and $\xi(t)$ is a white Gaussian noise $\langle\xi(t)\rangle = 0$, $\langle\xi(t)\xi(t')\rangle = \delta(t - t')$. We focus on one-dimensional (1D) Langevin dynamics representing a decision-

making process on the domain $x \in [-1; 1]$. In 1D, the force derives from the potential function $F(x) = -\mathrm{d}\Phi(x)/\mathrm{d}x$. The Langevin trajectories $x(t)$ are latent, i.e. only accessible through stochastic observations $Y(t)$. We work with observations that follow an inhomogeneous Poisson process with time-varying intensity $f(x(t))$ that depends on the latent trajectory $x(t)$ via a function $f(x)$ (Fig. 1a). Poisson noise models the variability of spike generation in a neuron.

The non-stationary data $Y(t)$ arise in non-equilibrium systems that perform computations. Such systems start their operation in a specific initial state and finish in a terminal state representing the outcome of the computation. The initial and terminal states are fundamentally different from the equilibrium state of the system. An example of such non-equilibrium computation is neural dynamics underlying perceptual decision making in the brain. Each decision process begins when a sensory stimulus is presented to a subject and terminates when the subject commits to a choice. Neural activity transiently evolves from the initial state at the stimulus onset until a choice is made, and different choices correspond to different terminal states of neural activity[25]. In experiment, multiple realizations of the decision process can be recorded under the same conditions, called trials. The statistics of trajectories $x(t)$ across trials differs from the steady-state distribution.

To model non-stationary dynamics, we introduce three components into our framework (Fig. 1a, b). First, $p_0(x)$ models

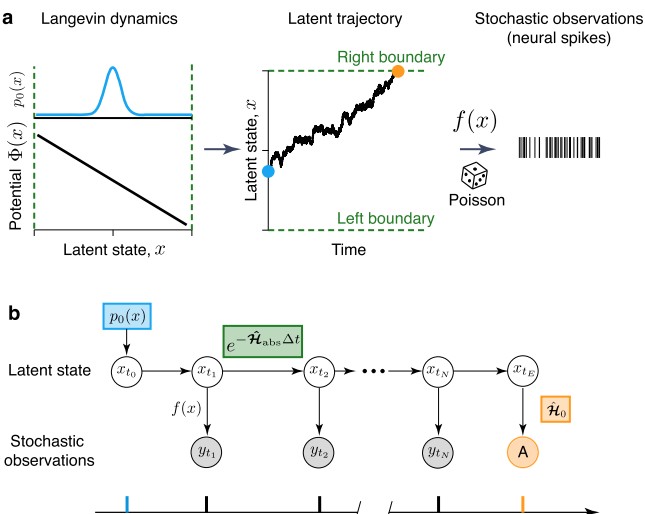

**Fig. 1 Inference framework for latent non-stationary Langevin dynamics. a** Latent dynamics are governed by the Langevin equation Eq. (1) with a deterministic potential $\Phi(x)$ and a Gaussian white noise with magnitude $D$. On each trial, the latent trajectory starts at the initial state $x(t_0)$ (blue dot) sampled from the probability density $p_0(x)$. When the trajectory reaches the domain boundaries (green dashed lines) for the first time, the observations can either terminate (orange dot) or continue depending on the experiment design. The latent Langevin dynamics are only accessible through stochastic observations, e.g., spikes that follow an inhomogeneous Poisson process with time-varying intensity that depends on the latent trajectory $x(t)$ via the firing-rate function $f(x)$. **b** Graphical diagram of the inference framework. Stochastic observations $y_{t_i}$ (gray circles) depend on the latent states $x_{t_i}$ (white circles), the arrows represent statistical dependencies. The absorption event (orange circle) indicates that observations terminate when the latent trajectory hit a boundary. The framework includes three non-stationary components: the initial state distribution $p_0(x)$ (blue box), the boundary conditions (reflecting or absorbing) for the time-propagation of latent dynamics (green box), and the absorption operator (orange box).

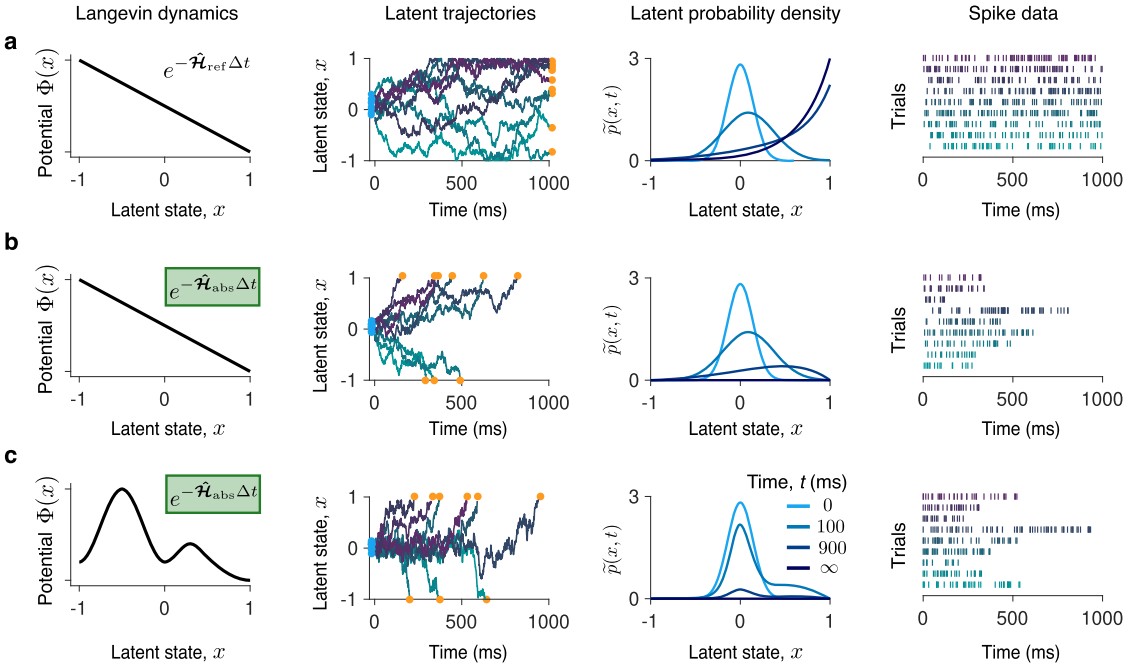

**Fig. 2 Observation noise masks qualitative differences in non-stationary Langevin dynamics.** Latent Langevin dynamics with: **a** a linear potential and reflecting boundaries; **b** a linear potential and absorbing boundaries; and **c** a non-linear potential and absorbing boundaries (first column). For each dynamics, nine example trajectories are displayed (color gradient, second column). In all cases, the initial latent state $x(t_0)$ (blue dots, second column) is sampled from the same density $p_0(x)$ (third column, $t = 0$). The time-propagation of the latent probability density $\widetilde{p}(x,t)$ strongly depends on the potential shape and boundary conditions (third column). With reflecting boundaries (**a**), the latent trajectories terminate anywhere in the latent space at the trial end, whereas with absorbing boundaries (**b**, **c**), the latent trajectories always terminate at the boundaries (orange dots, second column). These qualitative differences in the Langevin dynamics are difficult to discern from stochastic spike data (fourth column, colors correspond to the trajectories in the second column).

the distribution of the initial latent state at the trial start. On each trial, the latent dynamics evolve according to Eq. (1) from the initial condition $x(t_0) = x_0$, where $x_0$ is sampled from $p_0(x)$. The distribution $p_0(x)$ is latent and needs to be inferred from data. The two other components account for the mechanism terminating the observation on each trial, which can be controlled either by the experimenter or by the system itself. In decision-making experiments, these possibilities correspond to fixed-duration or reaction-time task designs[25]. In a fixed-duration task, the subject reports the choice after a fixed time period set by the experimenter. Even if the neural trajectory reaches a state representing a choice (i.e. a decision boundary) at an earlier time point, the deliberation process continues. Thus, the latent trajectory can terminate at any state at the trial end (Fig. 2a). In contrast, in a reaction-time task, the subject reports the choice as soon as the neural trajectory reaches a decision boundary for the first time. Thus trials have variable durations defined by the neural dynamics itself, and the latent trajectory always terminates at one of the decision boundaries at the trial end (Fig. 2b, c). To model these alternative scenarios, we impose appropriate boundary conditions for the Langevin dynamics Eq. (1): reflecting for the fixed-duration and absorbing for the reaction-time tasks. In addition, we derive an absorption operator enforcing the trajectory termination at a decision boundary in the reaction-time task (Fig. 1b).

The Poisson noise masks distinctions between different types of latent Langevin dynamics. The spike trains appear similar for dynamics with reflecting versus absorbing boundaries (Fig. 2a, b), and with a linear versus non-linear potential (Fig. 2b, c). A non-stationary initial state $p_0(x)$ is also not obvious in the spike trains. Distinguishing these qualitatively different dynamics based on

spike data is difficult, despite the latent trajectories and the corresponding time-dependent latent probability densities $\widetilde{p}(x,t)$ (Eq. (5) in Methods section) are different (Fig. 2).

We infer the force potential $\Phi(x)$, the noise magnitude $D$, and the initial distribution $p_0(x)$ from stochastic spike data $Y(t)$. The data consists of multiple trials $Y(t) = \{Y_i(t)\}$ ($i = 1, 2, \ldots n$), and for each trial $Y_i(t) = \{t_0^i, t_1^i, \ldots, t_{N_i}^i, t_E^i\}$, where $t_1^i, t_2^i, \ldots, t_{N_i}^i$ are recorded spike times, and $t_0^i$ and $t_E^i$ are the trial start and end times, respectively. We maximize the data likelihood $\mathscr{L}[Y(t)|\theta]$ with respect to $\theta = \{\Phi(x), p_0(x), D\}$. We derive analytical expressions for the variational derivatives of the negative log-likelihood, which we use to update $\theta$ using a gradient-descent (GD) algorithm[2] (Methods section). The variational derivatives of the potential $\Phi(x)$ and $p_0(x)$ are continuous functions, which we evaluate numerically on each GD step using a finite basis (Supplementary Note 1). Thus, our method is non-parametric in the sense that we do not specify a parametric form for the functions $\Phi(x)$ and $p_0(x)$, but use the analytical expressions for their continuous variational derivatives evaluated in a finite basis. The likelihood calculation involves time-propagation of the latent probability density with the operator $\exp(-\hat{\mathcal{H}}(t_i - t_{i-1}))$, where $\hat{\mathcal{H}}$ is a modified Fokker-Planck operator (Eq. (6) in Methods section). The operator $\hat{\mathcal{H}}$ satisfies either reflecting ($\hat{\mathcal{H}}_{\text{ref}}$, fixed-duration task), or absorbing boundary conditions ($\hat{\mathcal{H}}_{\text{abs}}$, reaction-time task). The absorbing boundary conditions ensure that trajectories reaching a boundary before the trial end do not contribute to the likelihood. In addition, the absorption operator $A$ enforces that the likelihood includes only trajectories terminating on the boundaries at the trial end time $t_E$ (Methods section).

**Contributions of non-stationary components to the accurate inference**. We found that accurate inference of non-stationary Langevin dynamics requires incorporating all three non-stationary components: the initial distribution $p_0(x)$, the boundary conditions, and the absorption operator. The necessity of all components for accurate inference is not obvious. Since spike trains generated from stationary versus non-stationary dynamics appear similar (Fig. 2), one could assume that omitting non-stationary components may affect the inference only insignificantly. To demonstrate how each component contributes to the accurate inference, we focus here on inferring the potential $\Phi(x)$ from synthetic data with known ground truth, assuming $p_0(x)$ and $D$ are provided (we consider simultaneous inference of $\Phi(x)$, $p_0(x)$, $D$ below). We use 200 trials of spike data generated from the model with a linear ground-truth potential and a narrow initial state distribution (full list of parameters in Supplementary Table 1). We simulated a reaction-time task, so that each trial terminates when the latent trajectory reaches one of the decision boundaries producing a non-stationary distribution of latent trajectories (Fig. 2b).

The inference accurately recovers the Langevin dynamics from these non-stationary spike data when all non-stationary components are taken into account (Fig. 3a). The GD algorithm iteratively increases the model likelihood (decreases the negative log-likelihood). Starting from an unspecific initial guess $\Phi(x) = \text{const}$, the potential shape changes gradually over the GD iterations. After some iterations, the fitted potential closely matches the ground-truth shape while the log-likelihood of the fitted model approaches the log-likelihood of the ground-truth model. The concurrent agreement of the inferred potential and its likelihood with the ground truth indicates the accurate recovery of the Langevin dynamics. At later iterations, the potential shape can deteriorate due to overfitting, and model selection is required

for identifying the model that accurately approximates dynamics in the data when the ground truth is not known[2] (we consider model selection and uncertainty quantification below).

To reveal how each non-stationary component contributes to the inference, we replace all components one by one with their stationary counterparts and evaluate the inference quality under these modified conditions. First, we test the importance of the absorption operator by performing the inference with the initial distribution $p_0(x)$ and absorbing boundary conditions, but omitting the absorption operator (Fig. 3b). In this scenario, the likelihood includes all trajectories that terminate anywhere in the latent space and do not reach the domain boundaries before the trial end. The inferred potential shows the correct linear slope, but develops a large barrier near the right boundary, where the ground-truth potential is low. This behavior arises since the spurious potential barrier reduces the probability flux through the absorbing boundary and hence increases the model likelihood. Accordingly, the likelihood is lower for the ground-truth potential than for the potential with the spurious barrier when the absorption operator is omitted in the likelihood calculation Eq. (4). The absorption operator corrects for this mismatch by ensuring that only trajectories terminating at the boundaries contribute to the likelihood.

Next, we test the importance of the absorbing boundary conditions using the same non-stationary data. We take into account the initial distribution $p_0(x)$, but replace the absorbing with reflecting boundary conditions in the inference (Fig. 3c). In this scenario, all trajectories contribute to the likelihood independent of when and whether they reach the domain boundaries. The inferred potential exhibits a small barrier near the right boundary where the ground-truth potential is low. The probability density of latent trajectories in the data is vanishing at the absorbing boundaries (Fig. 2b), whereas stationary dynamics

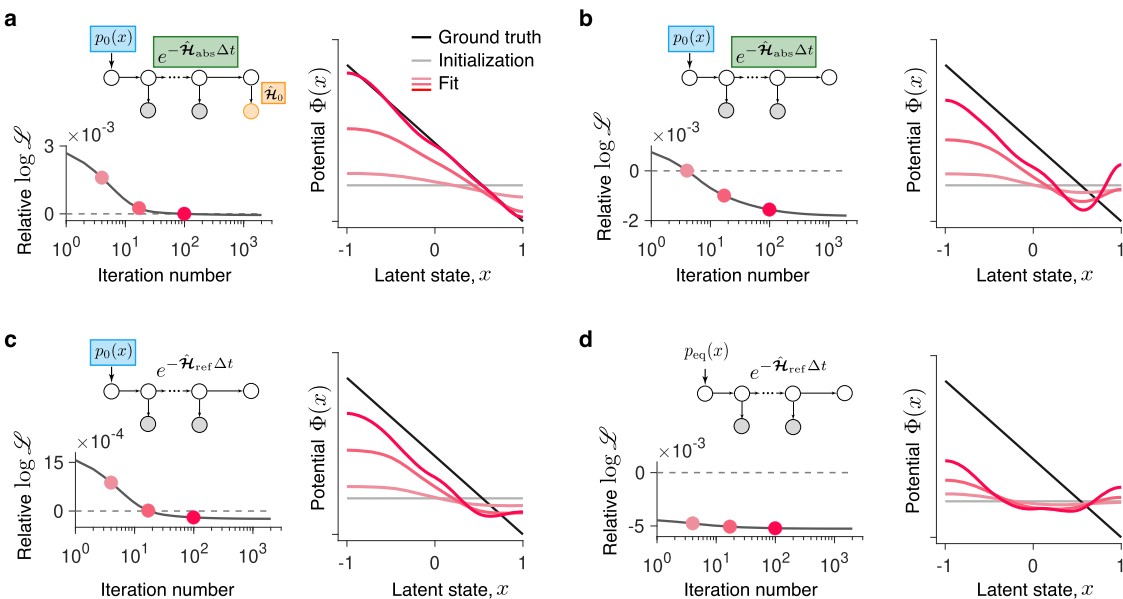

**Fig. 3 Contribution of non-stationary components to the accurate inference of latent Langevin dynamics.** The spike data are generated from the Langevin dynamics with a linear potential and absorbing boundaries (Fig. 2b). **a** The inference incorporates the non-equilibrium initial state distribution $p_0(x)$, absorbing boundary conditions, and the absorption operator (graphical diagram, inset in the left panel). When the likelihood of the fitted model approaches the likelihood of the ground-truth model (left panel, the relative log-likelihood is $\left[\log \mathscr{L}_{gt} - \log \mathscr{L}\right]/\log \mathscr{L}_{gt}$), the inferred potential shape closely matches the ground truth (right panel, colors correspond to the iterations marked with dots on the left panel). **b** Same as **a**, but omitting the absorption operator in the inference. The relative log-likelihood is with respect to the likelihood for the ground-truth potential, with the absorption operator omitted in the likelihood calculation for both the ground-truth and fitted potentials. **c** Same as **b**, but replacing the absorbing with reflecting boundary conditions in the inference and in the likelihood calculation for both the fitted and ground-truth potentials. **d** Same as **c**, but replacing $p_0(x)$ with the equilibrium density $p_{eq}(x)$ in the inference and in the likelihood calculation for both the fitted and ground-truth potentials. Omitting any of the non-stationary components results in artifacts in the inferred potentials.

with reflecting boundaries predict high probability density in the regions where the potential is low (Fig. 2a). Hence, the spurious potential barrier arises to explain the low probability density at the right boundary. Accordingly, the likelihood is higher for the potential with the spurious barrier than for the ground-truth potential when reflecting instead of absorbing boundary conditions are used in the likelihood calculation.

Finally, we test the importance of the initial state distribution $p_0(x)$. Using the same non-stationary data, we perform the inference with $p_0(x)$ replaced by the equilibrium distribution $p_{eq}(x) \propto \exp(-\Phi(x))$ under the reflecting boundary conditions (Fig. 3d). Instead of the linear slope, the inferred potential exhibits a flat shallow valley, which accounts for the high density of latent trajectories near the domain center in the data due to non-equilibrium $p_0(x)$. The equilibrium dynamics in the ground-truth potential predict lower probability density at the domain center than in the data, hence the likelihood is lower for the ground-truth potential than for the inferred shallow potential when incorrect initial distribution is used in the likelihood calculation. These results demonstrate that all three non-stationary components are critical for the accurate inference of non-stationary Langevin dynamics, and omitting any of them results in incorrect inference that accounts for the non-stationary data statistics by artifacts in the potential shape.

**Discovering models of decision-making.** To demonstrate that our framework can accurately infer qualitatively different non-stationary dynamics, we perform the inference on synthetic data generated by the alternative models of perceptual decision-making. We consider latent Langevin dynamics corresponding to the ramping and stepping models of decision making proposed previously[27]. The ramping model assumes that on single trials neural activity evolves gradually towards a decision boundary as a linear drift-diffusion process, which corresponds to a linear potential with a constant slope (Fig. 2b). The stepping model assumes that on single trials neural activity abruptly jumps from the initial to a final state representing a choice, which corresponds to a potential with two barriers where trajectories have to overcome one of the barriers to reach a decision boundary (Fig. 2c). Distinguishing between these alternative models of decision making is difficult with the traditional approach based on parametric model comparisons[28].

We generated spike data with the ramping and stepping latent dynamics in a reaction time task (Fig. 2b, c). We choose the potential $\Phi(x)$, noise magnitude $D$, and the initial state distribution $p_0(x)$ so that the speed and accuracy of decisions in the model is similar to typical experimental values, and $f(x)$ is chosen to produce realistic firing rates[31] (parameters provided in Supplementary Table 1). First, we infer the potential shape $\Phi(x)$ with the correct $D$ and $p_0(x)$ provided. For both ramping and stepping dynamics, our framework accurately infers the correct potential shape from 200 data trials (a realistic data amount in experiment, Fig. 4a, b). At the iteration when the likelihoods of the fitted and ground-truth model are equal, the inferred potentials are in good agreement with the ground truth, confirming the inference accuracy. The inference accuracy further improves with a larger data amount of 1600 trials.

Finally, we demonstrate simultaneous inference of all functions governing the non-stationary dynamics $\Phi(x)$, $p_0(x)$, and $D$ using synthetic data generated from the ramping model (Fig. 4c). We update each of $\Phi(x)$, $p_0(x)$, and $D$ in turn on successive GD iterations. As the likelihood of the fitted model approaches the likelihood of the ground-truth model, the potential shape, noise magnitude, and the initial state distribution all closely match the ground truth, confirming the accurate inference of a full model of

latent non-stationary Langevin dynamics. The inference of $p_0(x)$ can be less accurate when the data consist of a few long trials so that the dynamics equilibrate and trajectories contain little information about the initial state. In this quasi-stationary regime, $p_0(x)$ does not play an important role and the potential $\Phi(x)$ and noise magnitude $D$ that define equilibrium dynamics can be inferred accurately even with an inaccurate inference of $p_0(x)$ (Supplementary Fig. 1).

**Model selection and uncertainty quantification.** So far we validated the inference accuracy by comparing the fitted model and its likelihood with the ground truth. However, in practical applications, the ground truth is unknown, and we need a procedure for selecting the optimal model among many models produced across iterations of the gradient descent. On early iterations, the fitted models miss some features of the correct dynamics (underfitting), whereas on late iterations, the fitted models develop spurious features (overfitting). The model that best captures the correct dynamics is discovered at some intermediate iterations. The standard approach for selecting the optimal model is based on optimizing model's ability to predict new data (i.e. generalization accuracy), e.g., using cross-validation. However, methods optimizing generalization accuracy cannot reliably identify correct features and avoid spurious features when applied to flexible models[2]. An alternative approach for model selection is based on directly comparing features of the same complexity discovered from different data samples[2]. Since true features are the same, whereas noise is different across data samples, the consistency of features inferred from different data samples can separate the true features from noise. Model selection based on feature consistency can reliably identify the correct features for stationary dynamics[2]. Here we extended this method for the case of non-stationary dynamics (Methods section).

We illustrate the model selection method based on feature consistency using the same synthetic data as in Fig. 3a. We split the full dataset into two halves and optimize the model on each half independently. For each model produced by the gradient descent, we calculate the feature complexity defined as a negative entropy of latent trajectories $\mathcal{M} = -S[\Phi(x), D, p_0(x)]$ (Eq. (14)). For non-stationary dynamics, the entropy depends not only on the potential shape $\Phi(x)$ but also on the initial distribution $p_0(x)$ (Supplementary Note 6). The feature complexity grows over GD iterations as the model develops more and more structure (Fig. 5a). After the true features are discovered, $\mathcal{M}$ exceeds the ground-truth complexity, and further increases of $\mathcal{M}$ indicate fitting noise in the training data. The optimal feature complexity $\mathcal{M}^*$ separates the true features from noise. To determine the optimal $\mathcal{M}^*$ when the ground truth is unknown, we compare models of the same complexity discovered from two data halves. For $\mathcal{M} < \mathcal{M}^*$, the models of the same complexity tightly overlap between two data samples (Fig. 5c, left). For $\mathcal{M} > \mathcal{M}^*$, the models of the same complexity diverge, because overfitting patterns are unique for each data sample (Fig. 5c, right). We quantify the overlap of two models using the Jensen-Shannon divergence (JSD) between time-dependent probability densities of their latent trajectories (Eqs. (15) and (16)). For low feature complexities, JSD is small indicating that the true features of the dynamics are consistent between data samples (Fig. 5b, c, left). For higher feature complexities, JSD rises sharply indicating divergence of spurious features between data samples (Fig. 5b, c, right). We define the optimal $\mathcal{M}^*$ as the feature complexity for which JSD reaches a threshold. This procedure returns two overlapping potentials corresponding to $\mathcal{M}^*$, which agree with the ground-truth model on synthetic data (Fig. 5c, middle). We also performed model selection based on feature consistency for stepping and ramping

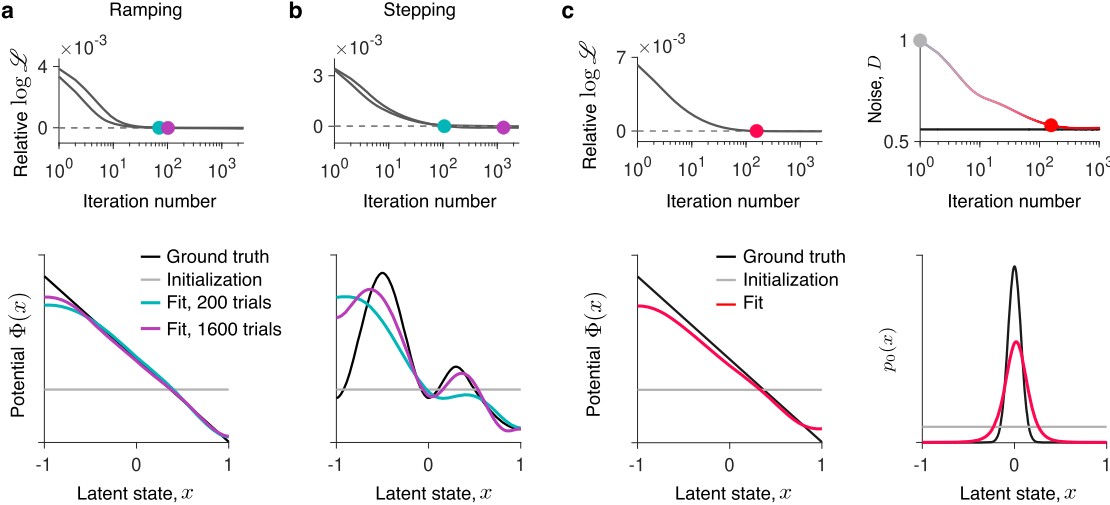

**Fig. 4 Inference of decision-making dynamics and simultaneous inference of all functions governing non-stationary Langevin dynamics. a** The spike data are generated from the Langevin dynamics with a linear potential and absorbing boundaries (Fig. 2b), which corresponds to the ramping model of decision-making dynamics. When the likelihoods of the fitted and ground-truth models are equal (upper panel, colored dots), the inferred potential closely matches the ground-truth potential (lower panel, colors correspond to dots in the upper panel). The inference accuracy improves with more data (teal - 200 trials, purple - 1600 trials). **b** Same as a, but for the spike data generated from the Langevin dynamics with a non-linear potential with two barriers and absorbing boundaries (Fig. 2c), which corresponds to the stepping model of decision-making dynamics. **c** Simultaneous inference of the potential $\Phi(x)$, the initial state distribution $p_0(x)$, and noise magnitude $D$ from the same spike data as in a (400 trials). As the likelihood of the fitted model approaches the likelihood of the ground-truth model (upper left), all fitted components simultaneously approach the ground truth.

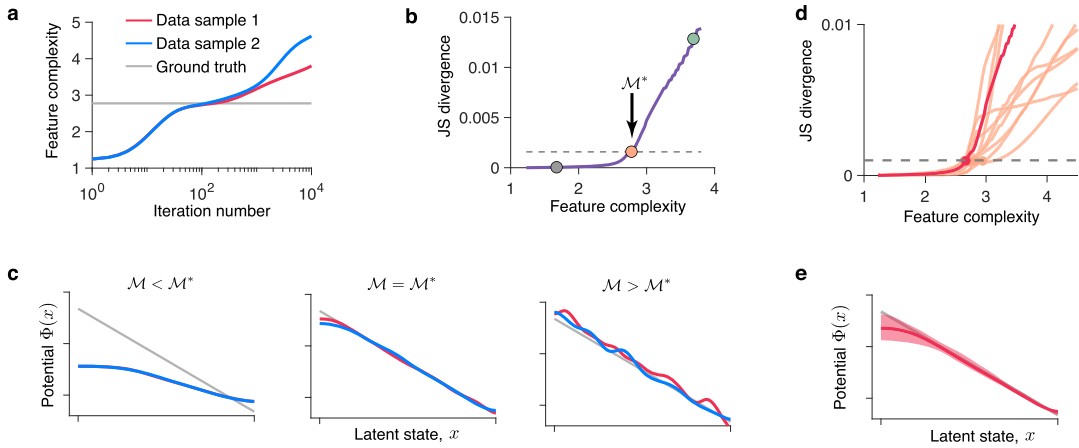

**Fig. 5 Model selection and uncertainty quantification. a** Feature complexity $\mathcal{M}$ increases over GD iterations at a rate that varies across data samples. The ground-truth feature complexity (gray line) is achieved on different iteration for different data samples. **b** JS divergence between the models discovered from data samples 1 and 2 for each level of the feature complexity. The optimal value $\mathcal{M}^*$ is defined as a maximum feature complexity for which JS divergence does not exceed a fixed threshold (dashed line). The dots correspond to the potential in **c**. **c** A pair of fitted potentials at $\mathcal{M} < \mathcal{M}^*$ (left, gray dot in **b**), $\mathcal{M} = \mathcal{M}^*$ (middle, orange dot in **b**), and $\mathcal{M} > \mathcal{M}^*$ (right, teal dot in **b**) for data samples 1 and 2 (colors correspond to data in **a**). In all panels the ground-truth model is shown in gray. **d** JS divergences between models of the same complexity discovered from two data halves for each of 10 bootstrap samples (red line - same data as in **b**). The same threshold is used to select a pair of optimal models for each bootstrap sample. **e** Fitted potential (red) is the average of two potentials at optimal $\mathcal{M}^*$ produced by the model selection (middle panel in **c**). The confidence bounds (shaded area) are obtained as a pointwise 5% and 95% percentile across twenty potentials produced by the model selection on 10 bootstrap samples in **d**. The ground-truth model is shown in gray.

dynamics (Fig. 4a, b) and found that the selected models agree well with the ground truth (Supplementary Fig. 2b, c).

Practical applications often require quantifying uncertainty of the inferred model, which can be performed via bootstrapping, as we illustrate using the same data as in Fig. 3a. To obtain confidence bounds for the inferred model, we generate ten bootstrap samples by sampling trials randomly with replacement from the set of all trials. For each bootstrap sample, we refit the model and perform model selection using our feature consistency method (Fig. 5d). We then obtain the confidence bounds for the inferred potential by

computing a pointwise 5% and 95% percentile across 20 potentials produced by the model selection on ten bootstrap samples (Fig. 5e). As expected, the uncertainty is largest in the regions where the potential is high, i.e. where the density of latent trajectories and hence the amount of spike data are low.

## Discussion

Our framework accurately infers non-stationary Langevin dynamics from stochastic observations and accommodates non-additive

Poisson observation noise. We demonstrate that accurate inference requires taking into account the non-equilibrium initial and final states that represent the start and outcome of computations performed by the system. Ignoring the non-equilibrium initial or final states results in incorrect inference, in which erroneous features in the dynamics arise due to non-stationary data distribution. We consider non-stationarity that arises from transient dynamics on a fast timescale within each trail, while the dynamical model components —potential, noise, and $p_0(x)$—are the same across trials. An additional source of non-stationarity may arise from slow drifts in the dynamics across trials[32,33]. Such non-stationary drifts can be modeled as changes in the potential, noise, and $p_0(x)$ on a slow timescale across trials.

The inference accuracy depends on the amount of available data and the complexity of underlying dynamics. Here we considered cases where the inference accuracy was not limited by the data amount to isolate how it depends on non-stationary components. A larger data amount generally results in more accurate inference (Fig. 4a, b). With insufficient data, the inference can underfit, i.e. not discover all features of the system's dynamics[2]. Inferring more complex dynamics requires larger amounts of data[2].

We illustrate our inference framework using models of neural dynamics during decision making, an inherently non-stationary process of transforming sensory information into a categorical choice. Comparisons between simple parametric models, which instantiate a discrete set of alternative hypotheses, proved ineffective to reveal the underlying neural dynamics[27–30]. An obvious pitfall is that none of the a priori guessed alternative hypotheses may be correct[28], and therefore model selection limited to a discrete set of hypotheses critically lacks flexibility. In contrast, our Langevin framework provides a flexible non-parametric description of dynamics, which covers a continuous space of hypotheses within a single model architecture[2]. Our framework can smoothly interpolate between many qualitatively different dynamics, which are all expressed with the same analytical equations, offering a powerful alternative to parametric model selection[28]. The inferred Langevin equation provides an interpretable description of dynamics, which opens access to many analytical tools available for the analysis, prediction, and control of stochastic systems[1,34]. Other approaches were also proposed to achieve the balance between flexibility and interpretability, for example, by approximating non-linear dynamics with a hierarchy of locally linear systems[35]. Flexible interpretable models can discover new hypotheses by fitting data, thus going beyond the classical model comparisons[2,36–38]. Our framework can be generalized to several latent dimensions and parallel data streams[39] (e.g., multi-neuron recordings) and opens new avenues for analyzing dynamics of complex systems far from equilibrium.

## Methods

### Maximum-likelihood inference of latent non-stationary Langevin dynamics.
We provide a brief summary of the analytical calculation of the model likelihood and its variational derivatives (see Supplementary Information for details). The likelihood $\mathscr{L}[Y(t)|\theta]$ is a conditional probability density of observing the data $Y(t)$ given a model $\theta = \{\Phi(x), p_0(x), D\}$. We only consider here a single trial $Y(t) = \{t_0, t_1, \ldots, t_N, t_E\}$, since the total data likelihood is a product of likelihoods of all trials. The likelihood $\mathscr{L}[Y(t)|\theta]$ is a probability density of the observed spike data, since in continuous time the probability of any precise spike sequence $\{t_1, t_2, \ldots t_N\}$ is infinitesimal. We can obtain the probability of observing a spike within $dt$ of each $\{t_1, t_2, \ldots t_N\}$ by multiplying the likelihood with $dt^N$.

The likelihood is obtained by marginalizing the joint probability density $P(\mathcal{X}(t), Y(t)|\theta)$ over all possible latent trajectories $\mathcal{X}(t)$ that may underlie the data[2,23]:

$$\mathscr{L}[Y(t)|\theta] = \int \mathscr{D}\mathcal{X}(t) \, P(\mathcal{X}(t), Y(t)|\theta). \tag{2}$$

Here $\mathcal{X}(t)$ is a continuous latent trajectory, and the path integral is performed over all possible trajectories. Note that if the trajectory $\mathcal{X}(t)$ was fixed and fully

observed, Eq. (2) would reduce to the well-known expression for the likelihood of an inhomogeneous Poisson process with the instantaneous firing rate $\lambda(t) = f(\mathcal{X}(t))$ (Supplementary Note 4). Since the latent trajectory that produced the data is unknown, we need to consider all possible latent paths weighted according to how consistent they are with the spike data and with the Langevin dynamics.

To compute the path integral in Eq. (2), we consider a disretized latent trajectory $X(t) = \{x_{t_0}, x_{t_1}, \ldots, x_{t_N}, x_{t_E}\}$, which is a discrete set of points along a continuous path $\mathcal{X}(t)$ at each of the observation times $\{t_0, t_1, \ldots, t_N, t_E\}$. Once we calculate the joint probability density $P(X(t), Y(t))$ of a discretized trajectory and data, then we can obtain the data likelihood by marginalization over all discretized latent trajectories:

$$\mathscr{L} = \int_{x_{t_0}} \int_{x_{t_1}} \cdots \int_{x_{t_N}} \int_{x_{t_E}} dx_{t_0} \cdots dx_{t_E} P(X(t), Y(t)). \tag{3}$$

Using the Markov property of the latent Langevin dynamics Eq. (1) and conditional independence of spike observations, the joint probability density $P(X(t), Y(t))$ can be factorized[24] (Fig. 1b):

$$P(X(t), Y(t)) = p(x_{t_0})\left(\prod_{i=1}^{N} p(y_{t_i}|x_{t_i})p(x_{t_i}|x_{t_{i-1}})\right)p(x_{t_E}|x_{t_N})p(A|x_{t_E}). \tag{4}$$

Here $p(y_{t_i}|x_{t_i})dt$ is the probability of observing a spike within small $dt$ of time $t_i$ given the latent state $x_{t_i}$, hence $p(y_{t_i}|x_{t_i}) = f(x_{t_i})$ by the definition of the instantaneous Poisson firing rate. $p(x_{t_0})$ is the probability density of the initial latent state. $p(x_{t_i}|x_{t_{i-1}})$ is the transition probability density from $x_{t_{i-1}}$ to $x_{t_i}$ during the time interval between the adjacent spike observations, which accounts for the absence of spikes during this time interval. Finally, the term $p(A|x_{t_E})$ represents the absorption operator, which ensures that only trajectories terminating at one of the domain boundaries at time $t_E$ contribute to the likelihood. The absorption term $p(A|x_{t_E})$ is only applied in the case of absorbing boundaries, and it is absent in the case of reflecting boundaries (Supplementary Note 1).

The discretized latent trajectory $X(t) = \{x_{t_0}, x_{t_1}, \ldots, x_{t_N}, x_{t_E}\}$ is obtained by marginalizing the continuous trajectory $\mathcal{X}(t)$ over all latent paths connecting $x_{t_{i-1}}$ and $x_{t_i}$ during each interspike interval. These marginalizations are implicit in the transition probability densities. For the Langevin dynamics Eq. (1), the time-dependent probability density $\tilde{p}(x, t)$ evolves according to the Fokker-Planck equation[34]:

$$\frac{\partial \tilde{p}(x, t)}{\partial t} = \left(-D\frac{\partial}{\partial x}F(x) + D\frac{\partial^2}{\partial x^2}\right)\tilde{p}(x, t) \equiv -\hat{\mathcal{H}}_0\tilde{p}(x, t), \tag{5}$$

which accounts for the drift and diffusion in the latent space (Fig. 2, third column). In addition, the transition probability density $p(x_{t_i}|x_{t_{i-1}})$ in Eq. (4) should also account for the absence of spike observations during intervals between adjacent spikes in the data. Thus, $p(x_{t_i}|x_{t_{i-1}})$ satisfies a modified Fokker-Planck equation (Supplementary Note 5):

$$\frac{\partial p(x, t)}{\partial t} = \left(-D\frac{\partial}{\partial x}F(x) + D\frac{\partial^2}{\partial x^2} - f(x)\right)p(x, t) \equiv -\hat{\mathcal{H}}p(x, t), \tag{6}$$

where the term $-f(x)$ accounts for the probability decay due to spike emissions[2,23]. The solution of this equation $p(x, t_i) = p(x, t_{i-1})\exp(-\hat{\mathcal{H}}(t_i - t_{i-1}))$ propagates the latent probability density forward in time during each interspike interval. Depending on the experiment design, we solve Eq. (6) with either absorbing or reflecting boundary conditions (Supplementary Note 1).

The term $p(A|x_{t_E})$ in Eq. (4) represents the absorption operator $A$, which ensures that the likelihood only includes trajectories terminating at the boundaries. The instantaneous probability $p_A$ for a trajectory to be absorbed at the boundaries given the latent state $x_{t_E}$ is obtained by applying $A$ to a delta-function initial condition $\delta(x_{t_e})$ and then integrating over the latent space:

$$p_A = \int_{-1}^{1} \delta(x_{t_e})A dx. \tag{7}$$

To derive the absorption operator, we consider the survival probability $P_{\Delta t}(S_{t_E}|x_{t_E})$ for a trajectory to survive (i.e. not to be absorbed at the boundary) within a time interval $\Delta t$ given the latent state $x_{t_E}$. The survival probability is obtained by propagating the initial condition $\delta(x_{t_E})$ with the operator $\exp(-\hat{\mathcal{H}}_0\Delta t)$ and integrating the result over the latent space:

$$P_{\Delta t}(S_{t_E}|x_{t_E}) = \int_{-1}^{1} \delta(x_{t_E})\exp(-\hat{\mathcal{H}}_0\Delta t)dx. \tag{8}$$

Here we use the operator $\hat{\mathcal{H}}_0$ instead of the operator $\hat{\mathcal{H}}$, because the survival probability accounts only for the probability loss due to absorption at the boundaries and not for the probability decay due to spike emissions.

The probability for a trajectory to be absorbed during a time interval $\Delta t$ given the state $x_{t_E}$ is given by $P_{\Delta t}(A_{t_E}|x_{t_E}) = 1 - P_{\Delta t}(S_{t_E}|x_{t_E})$. Thus, the instantaneous

probability of absorption is obtained as

$$p_A = \lim_{\Delta t \to 0} \frac{P_{\Delta t}(A_{t_E}|x_{t_E})}{\Delta t} = \lim_{\Delta t \to 0} \frac{1 - P_{\Delta t}(S_{t_E}|x_{t_E})}{\Delta t} = \int_{-1}^{1} dx \delta(x_{t_E}) \hat{\mathcal{H}}_0, \quad (9)$$

where we use Eq. (8) to take the limit. Comparing this result with Eq. (7), we find that $A = \hat{\mathcal{H}}_0$. Note that $-\hat{\mathcal{H}}_0$ is the Fokker-Planck operator in Eq. (5) that describes the rate of change of the latent probability density at each location $x$. Integrating both sides of Eq. (5) over the latent space, we obtain

$$\frac{d\widetilde{p}(t)}{dt} = \frac{d}{dt} \int_{-1}^{1} dx \widetilde{p}(x,t) = -\int_{-1}^{1} dx A \widetilde{p}(x,t). \quad (10)$$

This equation describes the decay of the total probability $\widetilde{p}(t) = \int_x dx \widetilde{p}(x,t)$ in the latent space due to probability flux through the absorbing boundaries. Thus, applying the absorption operator $A$ and integrating over the latent space represents the instantaneous loss of the total probability at time $t$, which is the fraction of all survived trajectories that reach the absorbing boundaries at exactly time $t$.

To compute and optimize the likelihood numerically, we represent Eq. (3) in a discrete basis[39] (Supplementary Note 1). In the discrete basis, all continuous functions, such as $p_0(x)$, are represented by vectors, and the transition, emission, and absorption operators are represented by matrices. Thus, Eq. (3) is evaluated as a chain of vector-matrix multiplications.

**Gradient descent optimization**. We minimize the negative log-likelihood with the gradient descent (GD) algorithm. Instead of directly updating the functions $\Phi(x)$ and $p_0(x)$, we, respectively, update the driving force $F(x) = -\Phi'(x)$ and an auxiliary function $F_0(x) \equiv p'(x)/p_0(x)$. The potential $\Phi(x)$ and $p_0(x)$ are obtained from $F(x)$ and $F_0(x)$ via

$$\Phi(x) = -\int_{-1}^{x} F(s)ds + C, \quad p_0(x) = \frac{\exp\left(\int_{-1}^{x} F_0(s)ds\right)}{\int_{-1}^{1} \exp\left(\int_{-1}^{s'} F_0(s)ds\right)ds'}. \quad (11)$$

We fix the arbitrary additive constant $C$ in the potential to satisfy $\int_x \exp[-\Phi(x)]dx = 1$. The change of variable from $p_0(x)$ to $F_0(x)$ allows us to perform an unconstrained optimization of $F_0(x)$, and Eq. (11) ensures that $p_0(x)$ satisfies the normalization condition for a probability density $\int_{-1}^{1} p_0(x)dx = 1$, $p_0(x) \geqslant 0$. We ensure the positiveness of the noise magnitude $D$ by rectifying its value after each GD update $D = \max(D, 0)$.

We derive analytical expressions for the variational derivatives of the likelihood $\delta\mathcal{L}/\delta F(x)$, $\delta\mathcal{L}/\delta F_0(x)$ and the derivative $\partial\mathcal{L}/\partial D$, which are then evaluated in the discrete basis (Supplementary Notes 2 and 3). On each GD iteration, we update the model by stepping in the direction of the log-likelihood gradient:

$$\Theta_{n+1} = \Theta_n + \gamma_\Theta \frac{1}{\mathcal{L}} \frac{\delta\mathcal{L}}{\delta\Theta}. \quad (12)$$

Here $\Theta$ is one of the functions $F(x)$, $F_0(x)$, or the noise magnitude parameter $D$, with the corresponding learning rates $\gamma_\Theta > 0$, and $n$ is the iteration number. For simultaneous inference of $\Phi(x)$, $p_0(x)$, and $D$ (Fig. 4c), we update each of $F(x)$, $F_0(x)$, and $D$ in turn on successive GD iterations. The list of optimization hyperparameters, including learning rates and initializations, is provided in Supplementary Table 1.

**Synthetic data generation**. To generate synthetic spike data from a model with given $\Phi(x)$, $p_0(x)$, and $D$, we numerically integrate Eq. (1) with the Euler-Maruyama method to produce latent trajectories $x(t)$ on each trial. We then use time-rescaling method[40] to generate spike times from an inhomogeneous Poisson process with the firing rate $\lambda(t) = f(x(t))$. We use 200 data trials in Fig. 3; 200 and 1600 trials in Fig. 4a, b; and 400 trials in Fig. 4c. These data amounts are typical for experiments in which neural activity is recorded during decision making[25,31]. A single experimental session usually contains a total of 1000−2000 trails under different behavioral conditions, with about 100−200 trials in each condition.

**Feature complexity**. We define feature complexity as the negative entropy of latent trajectories generated by the model[2] $\mathcal{M} = -S[\Phi(x), D, p_0(x); \Phi^R(x), D^R, p_0^R(x)]$. The trajectory entropy is defined as a negative Kullback-Leibler (KL) divergence between the distributions $P[\mathcal{X}(t)]$ and $Q[\mathcal{X}(t)]$[41]:

$$S[\Phi(x), D, p_0(x); \Phi^R(x), D^R, p_0^R(x)] = -\int_0^{t_{obs}} \mathscr{D}\mathcal{X}(t) P[\mathcal{X}(t)] \ln \frac{P[\mathcal{X}(t)]}{Q[\mathcal{X}(t)]}. \quad (13)$$

$P[\mathcal{X}(t)]$ is the distribution of trajectories in the model of interest with Langevin parameters $\{\Phi(x), D, p_0(x)\}$, and $Q[\mathcal{X}(t)]$ is the distribution of trajectories in the reference model with Langevin parameters $\{\Phi^R(x), D^R, p_0^R(x)\}$. The path integral is performed over all possible trajectories $\mathcal{X}(t)$. The reference model is a free diffusion with zero driving force (i.e. constant potential $\Phi^R(x) = $ const) and the same diffusion coefficient $D$ as in the model of interest. The analytical expression for the trajectory entropy was derived previously for equilibrium dynamics[41]. In the equilibrium case, the trajectory entropy is defined by the equilibrium distribution $p_{eq}(x)$ and does not depend on the initial state distribution $p_0(x)$. We generalized this result and derived an expression for the trajectory entropy for non-stationary dynamics (Supplementary Note 6):

$$S[\Phi(x), D, p_0(x); \Phi^R(x), D, p_0^R(x)] = -\int dx p_0(x) \ln \frac{p_0(x)}{p_0^R(x)} - \frac{D}{4} \int_0^\infty dt \int dx F^2(x) p(x,t). \quad (14)$$

For non-stationary dynamics, the trajectory entropy depends on the time-dependent distribution of the latent trajectories $p(x,t)$ and thus on $p_0(x)$. We choose the initial distribution $p_0^R(x)$ for the reference model to be uniform. We derived an expression for efficient numerical evaluation of Eq. (14), where we take the integral over time analytically in the eigenbasis of the operator $\mathcal{H}_0$ (Supplementary Eq. (69) in Supplementary Note 6).

**Model selection based on feature consistency**. To select the optimal model, we compare models discovered from two non-intersecting halves of the data and evaluate the consistency of their features. We quantify the overlap between two models by evaluating Jensen-Shannon divergence (JSD) between their time-dependent probability densities over the latent space:

$$D_{JS} = \int_0^\infty JSD(\hat{p}^1(x,t)||\hat{p}^2(x,t))dt, \quad (15)$$

where

$$JSD(p(x)||q(x)) = \frac{1}{2}\left(\int p(x)\ln\frac{2p(x)}{p(x)+q(x)}dx + \int q(x)\ln\frac{2q(x)}{p(x)+q(x)}dx\right). \quad (16)$$

The probability density $\hat{p}(x,t)$ is normalized to account for the probability loss through the absorbing boundaries: $\hat{p}(x,t) = \widetilde{p}(x,t) + I_p\delta(x - x_b)$. Here $\widetilde{p}(x,t)$ is the time-dependent solution of Eq. (5), $I_p = 1 - \int \widetilde{p}(x,t)dx$ is the total probability loss through the absorbing boundaries up to time $t$, and $x_b$ denotes a boundary (we combine the probability loss through both boundaries into a single term). We approximate the time integral in Eq. (15) by the midpoint rule and calculate the time-dependent probability densities by numerically solving the Fokker-Planck Eq. (5) (Supplementary Note 6.3).

We compare models of roughly the same complexity between the two sets of models $\{\Phi_n^1(x), D^1, p_0^1(x)\}$ and $\{\Phi_n^2(x), D^2, p_0^2(x)\}$ produced by GD on each data half ($n = 1, 2, ...N$ is the iteration number). First, we calculate the feature complexities $\mathcal{M}_n^1$ and $\mathcal{M}_n^2$ for all models in the two sets. Since feature complexities do not match exactly between the two sets due to nuances in the data, we need to allow for some slack in feature complexity when comparing models[2]. Accordingly, for each level of feature complexity $\mathcal{M}_i^1$, we find the index $j^*$ that minimizes the absolute difference $|\mathcal{M}_i^1 - \mathcal{M}_{j^*}^2|$. Next, we calculate $D_{JS}(j)$ between the model with feature complexity $\mathcal{M}_i^1$ and each of the models in the second set with similar complexity $\mathcal{M}_j^2$, where $j = j^* - R, j^* - R + 1, ...j^* + R$ and we set $R = 5$. We then set $D_{JS}(\mathcal{M}_i^1) = \min_j D_{JS}(j)$. We repeat this procedure for different iterations $i$ to obtain the dependence $D_{JS}(\mathcal{M})$ (Fig. 5b, d). To find the optimal feature complexity, we set the threshold $D_{JS,thres} = 0.001$ and select $\mathcal{M}^*$ as the maximum feature complexity for which $D_{JS} \leqslant D_{JS,thres}$. This procedure returns two overlapping potentials of roughly the same feature complexity which represent the consistent features of dynamics across data samples (Fig. 5c).

## Data availability
The synthetic data used in this study can be reproduced using the source code.

## Code availability
The source code to reproduce the results of this study is freely available on GitHub (https://github.com/engellab/neuralflow, https://doi.org/10.5281/zenodo.5512552).

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

## Acknowledgements

This work was supported by the NIH grant R01 EB026949 (T.A.E. and M.G.), the Swartz Foundation (M.G.), and Katya H. Davey Fellowship (O.H.). We thank C. Aghamo-hammadi for thoughtful comments on the manuscript.

## Author contributions

M.G. and T.A.E. designed the research and developed the framework. M.G. and O.H. developed the code and performed computer simulations. M.G. and T.A.E. wrote the paper with input from O.H.

## Competing interests

The authors declare no competing interests.
