## [Peer Review File · Nature Communications]

Reviewer #1:
Remarks to the Author:

This paper presented a parameter estimation method for non-stationary Langevin equation with stochastic observations. The authors developed a max likelihood approach to determine parameters in the dynamics and the initial density. Overall, I think this is a nice contribution to addressing an important problem that comes up in various areas. But there are several important issues that the author should address.

1. On page 4, the article defined the functions $\Phi(x)$ and $p_0(x)$ as part of the entire parameter set. The 'space of continuous functions' described in the same sentence is in principle infinite dimensional. There is no way to infer these functions if the problem is placed in such a general setting. It is important to distinguish parametric and non-parametric methods before an estimation method is put forward. If a parametric method is pursued, which I believe is what this paper is doing, the effort should be confined to specific function classes with finitely many parameters.
2. Regarding the estimation of the drift, the most commonly used method is based on the Girsanov theorem. It should work in both stationary and non-stationary settings. This standard method should be mentioned in the introduction.
3. My first concern is with the estimation of $p_0(x)$. The problem itself is ill-posed. This can be seen from the underlying Fokker-Planck equation, which is of parabolic type and solving it backward in time is known to be ill-posed: a slight perturbation can drastically change the solution. The introduction of $f(x)$ actually makes the problem worse. One scenario when the initial configuration can be inferred with robust accuracy is if the observations stay very close to $t = 0$.
4. The potential/force considered in this paper is a bit too simple: 1-dimensional and linear. It is not clear whether this can be applied to more practical situations. For instance, in general, there is no explicit formula for the transition density of the Fokker-Planck equation.

Reviewer #2:

Remarks to the Author:

The authors present an extension to their model of flexible latent neural dynamics. These new extensions make the method more appropriate for fitting trial-based data by including trial start and end conditions on the dynamics. Although no real data applications are shown, trial-based experiments are extremely common in neuroscience, and thus these additions greatly extend the use of this modeling framework beyond the authors' previous work. Inference of these model components is demonstrated using a few simulations. However, I have several major concerns about the technical rigor and completeness of the manuscript.

1. My biggest concern is that the setup for the solution to the likelihood is not strictly correct (page 8, equation 3; or eq 14 in Haas et al 2013) and is very difficult to read. The factorization of probabilities as written does not make sense for a Poisson process: notably that $p(y_t | x_t)$ is the probability of a spike time, not a Poisson random variable for a bin. Thus, this isn't a well-defined probability (the probability of a spike at any time is 0). This derivation would be both more clear and more correct to write out the likelihood of the Poisson process instead of that factorization: $P(X, Y) \propto \exp(-\int_0^T f(x_s) ds) \prod_{t \in \text{spk times}} \lambda(x_t) p(X)$

Marginalizing this over X gives

$$P(Y) \propto E_{\{P(X)\}} \left[\exp(-\int_0^T f(x_s) ds) \prod_{t \in \text{spk times}} \lambda(x_t) \right] \\ = E_{\{P(X)\}} \left[\exp(-\int_0^T (f(x_s) - \sum_{t \in \text{spk times}} \delta_t(s) \log(\lambda(x_t))) ds) \right]$$

Where I'm sloppily using the δ as the Dirac delta to account for the intensity at spike times. My understanding is that your method uses the fact that the Feynman-Kac formula solves these path integrals as the solution of a PDE. The observation operator you use gives the $\lambda(\text{spike time})$ terms; therefore, even though eq 3 isn't quite right, your solution of the likelihood looks okay to me.

More explanation is needed about how your method computes this integral, as the Feynman-Kac is based on the Kolmogorov backward equation, but you are solving the forward equation (Fokker-Planck). Otherwise, this method is presented too opaquely for the target neuroscience audience. By writing out the log likelihood and the integral you're solving, it would more clearly link this method to the existing literature on Poisson processes in neuroscience or the broader statistical literature on Cox processes to your method (eq 1 in Paninski, 2004 for the log likelihood).

2. The conclusions in the 1st paragraph of page 5 ("To reveal how each non-stationary component contributes to the inference...") show a potentially serious challenge to the applicability of the model. Most importantly, I did not understand the authors' explanation for how the log likelihood of the fitted model is better than ground truth in Fig 3b. If this is really the case, it's unclear how or if this method could be used for model comparison/selection. As part of this, the distinction between the absorption operator and the absorbing boundary conditions needs to be explained more here ("the importance of the absorption operator by performing the inference with the initial distribution $p_0(x)$ and absorbing boundary conditions, but omitting the absorption operator").

3. The limitations of the current method to address model comparison challenges appear overstated. On page 7, the authors state that "comparisons between simple parametric models proved ineffective to reveal the underlying neural dynamics." This paper, however, demonstrates how modeling choices still exist even in more flexible approaches: does one include an absorbing bound? What is the dimensionality of the latent state? Are the dynamics constant across trials? Furthermore, the methods reported here do not report uncertainty in the model fit (many potentials explain the data similarly, even in simulation) or provide ways to quantitatively ask how well the dynamics in the data reflect different hypotheses (as comparing simple parametric models is designed to do). Thus, I think the authors need to be more exact and tone down how much this approach can do to solve the model comparison/selection problem.

4. The optimization process used to fit the model needs to be better defined. The plots show the log likelihood trajectory and optimization at a few epochs. It's not clear why these epochs were chosen – and a well-defined stopping criterion ought to be applied (not just that the term looked good at certain epochs). Additionally, it's not clear what the purpose of showing the optimization

at several epochs is in the current paper. In the author's previous paper, they proposed an early stopping method for their model, and thus it made more sense to show the gradient decent process. Here, it's confusing to see several suboptimal models, and I would suggest removing them. Lastly, the authors state on page 5 that model selection is needed to identify "the correct model when the ground truth is not known". No "correct" model exists in real data, and I would suggest rewording this to make it more clear how fitting should be approach when using real data.

5. The use of the word "nonstationary" throughout the paper was, in my opinion, often a poor word choice. The new terms being estimated appear to be constant across trials: the draws from the initial state distribution are different across trials, but that distribution itself is the same. I think it would be more clear to label these as sources of trial-to-trial variability instead of non-stationarity to make it clear that the potential (or other terms) doesn't depend on time. (Non-stationary terms could be, for instance, gain changes in firing rate that occur slowly over trials).

6. The authors should address related methodology for estimating single-trial dynamics and possibly present a comparison to one or more other approaches (e.g., Nassar et al. "Learning Structured Neural Dynamics From Single Trial Population Recording." 2018 52nd Asilomar Conference on Signals, Systems, and Computers. IEEE, 2018.)

7. The rigorousness of the approach could be improved by providing some limits on the convergence of the estimator (what can be estimated). The current paper suggests that any continuous potential can be fit, but without proof this feels too good to be true. What kinds of functions $\Phi(x)$ can be estimated given the uncertainty of how x can be observed through sparse spiking? I realize that strong proofs are beyond the scope of this paper, but I would like some guidelines for realistic inference - even really handwavy ones - to recognize this problem (e.g., if Φ isn't too wiggly, as was the case in your examples).

Minor points

1. Page 7, first paragraph: "Our framework ... can accommodate different observation processes, e.g., non-additive Poisson observation noise." Suggest rewording to say it accommodates Poisson observations as the integration approach here strongly depends on it being Poisson and no other noise models have been demonstrated.

2. Page 7, first paragraph: "Our framework generalizes to several latent dimensions". Given that this is the second paper the authors have written on this method, it would be good to see a 2D latent example to show that this claim is practical, not just theoretical.

Reviewer #3:
None

Reviewer #1 (Remarks to the Author):

This paper presented a parameter estimation method for non-stationary Langevin equation with stochastic observations. The authors developed a max likelihood approach to determine parameters in the dynamics and the initial density. Overall, I think this is a nice contribution to addressing an important problem that comes up in various areas. But there are several important issues that the author should address.

Reply:

We thank the reviewer for this generous assessment and for many insightful comments that help us improve the paper.

1) On page 4, the article defined the functions $\Phi(x)$ and $p_0(x)$ as part of the entire parameter set. The 'space of continuous functions' described in the same sentence is in principle infinite dimensional. There is no way to infer these functions if the problem is placed in such a general setting. It is important to distinguish parametric and non-parameter methods before an estimation method is put forward. If a parametric method is pursued, which I believe is what this paper is doing, the effort should be confined to specific function classes with finitely many parameters.

Reply:

Our analytical derivations of the likelihood and its variational derivatives are carried out in terms of continuous functions $\Phi(x)$ and $p_0(x)$ without using any finite bases, see our result Eq. (S10) and Supplementary Information 4. For numerical optimization, we evaluate these analytical expressions in a finite basis, as described in Supplementary Information 1. However, we do not assume any specific parametric form for these functions. Thus, our method is non-parametric in a similar sense as the histogram, kernel, or nearest-neighbor methods for density estimation are considered non-parametric in the machine-learning literature (Bishop 2006, chapter 2). We clarified this point on page 4 in the revised manuscript.

2) Regarding the estimation of the drift, the most commonly used method is based on the Girsanov theorem. It should work in both stationary and non-stationary setting. This standard method should be mentioned in the introduction.

Reply:

We were able to find some theoretical work that derives a maximum likelihood estimator for a parameter of the drift term using Girsanov theorem. We cited two papers on this topic in the revised Introduction: Wei & Shu (2015) and Wei (2018). Note that this method assumes a known parametric form for the drift, unlike our non-parametric method.

3) My first concern is with the estimation of $p_0(x)$. The problem itself is ill-posed. This can be seen from the underlying Fokker-Planck equation, which is of parabolic type and solving it backward in time is known to be ill-posed: a slight perturbation can drastically change the solution. The introduction of $f(x)$ actually makes the problem worse. One scenario when the initial configuration can be inferred with robust accuracy is if the observations stay very close to $t = 0$.

Reply:

The reviewer is right that solving the backward Fokker-Planck equation to find the initial state (in the absence of intermediate observations) is an ill-posed problem. However, this is not what we do. First, spike observations provide information about the entire latent trajectory from the initial to terminal state. Second, the analytical expressions for the likelihood and its variational derivatives involve only a solution of the forward Fokker-Planck equation between adjacent spike times, see Eq. (6). The corresponding backward equation does not appear in our analytical results, and we do not solve it numerically.

The reviewer is correct that the accuracy of inferring the initial state distribution $p_0(x)$ depends on the trial durations. However, this dependence is due to the amount of information about the initial state available in the data, and not due to the ill-posed problem of solving the backward equation (which we never do). The initial state influences the trajectory only at short times before the system equilibrates. If the data consists of a few long trials such that the dynamics are largely at equilibrium, then these data has little information about the initial state and the inference of $p_0(x)$ is less accurate. Nevertheless, even with an inaccurate estimate of $p_0(x)$, the potential and noise magnitude can be inferred accurately, as this information is contained in the long trajectory. We considered this stationary case in our previous work (Genkin & Engel, 2020), in which we simply assumed an equilibrium distribution for the initial state. In contrast, our current work focuses specifically on inferring transient non-stationary dynamics, when trials are short so that the system cannot equilibrate and the information about the initial state is contained in the trajectories. To clarify this point, we added a new Supplementary Fig. 1 and a discussion on page 7 in the revised manuscript.

4) The potential/force considered in this paper is a bit too simple: 1-dimensional and linear. It is not clear whether this can be applied to more practical situations. For instance, in general, there is not explicit formula for the transition density of the Fokker-Planck equation.

Reply:

We would like to clarify that our framework does not assume a parametric form for the potential function and the inferred potential can take arbitrary shape. Besides the example of a linear potential, we demonstrate the inference of a non-linear potential with two barriers (Fig. 4b), which can be approximated by a 14-degree polynomial (see Supplementary Table 1), and of a double-well potential (Supplementary Fig. 1). Our inference method does not rely on explicit formulas for the transition probability density. We solve the Fokker-Planck equation Eq. (6) numerically, which allows us to calculate the transition probability density for arbitrary potential shapes. Our previous work (Genkin & Engel 2020) provides more examples of inferring complex potential shapes in stationary case.

Reviewer #2 (Remarks to the Author):

The authors present an extension to their model of flexible latent neural dynamics. These new extensions make the method more appropriate for fitting trial-based data by including trial start and end conditions on the dynamics. Although no real data applications are shown, trial-based experiments are extremely common in neuroscience, and thus these additions greatly extend the use of this modeling framework beyond the authors' previous work. Inference of these model components is demonstrated using a few simulations. However, I have several major concerns about the technical rigor and completeness of the manuscript.

Reply:

We thank the reviewer for the overall positive assessment and suggestions that helped us improve the paper.

1) My biggest concern is that the setup for the solution to the likelihood is not strictly correct (page 8, equation 3; or eq 14 in Haas et al 2013) and is very difficult to read. The factorization of probabilities as written does not make sense for a Poisson process: notably that $p(y_t | x_t)$ is the probability of a spike time, not a Poisson random variable for a bin. Thus, this isn't a well-defined probability (the probability of a spike at any time is 0). This derivation would be both more clear and more correct to write out the likelihood of the Poisson process instead of that factorization:

$$P(X, Y) \propto \exp(-\int_0^T f(x_s) ds) \prod_t \text{in } \text{trm}\{\text{spk times}\} \lambda(x_t) p(X)$$

Marginalizing this over X gives

$$P(Y) \propto \int P(X) \left[\exp(-\int_0^T f(x_s) ds) \prod_t \text{in } \text{trm}\{\text{spk times}\} \lambda(x_t) \right]$$

$$= \int P(X) \left[\exp(-\int_0^T (f(x_s) - \sum_t \text{in } \text{trm}\{\text{spk times}\} \delta_t(s) \log(\lambda(x_t))) ds) \right]$$

Where I'm sloppily using the δ as the Dirac delta to account for the intensity at spike times. My understanding is that your method uses the fact that the Feynman-Kac formula solves these path integrals as the solution of a PDE. The observation operator you use gives the $\lambda(\text{spike time})$ terms; therefore, even though eq 3 isn't quite right, your solution of the likelihood looks okay to me. More explanation is needed about how your method computes this integral, as the Feynman-Kac is based on the Kolmogorov backward equation, but you are solving the forward equation (Fokker-Planck). Otherwise, this method is presented too opaquely for the target neuroscience audience. By writing out the log likelihood and the integral you're solving, it would more clearly link this method to the existing literature on Poisson processes in neuroscience or the broader statistical literature on Cox processes to your method (eq 1 in Paninski, 2004 for the log likelihood).

Reply:

We thank the reviewer for pointing out that the description of our methods required additional clarification. To address this issue, we revised Methods in the main paper and added two new sections to Supplementary Information. In Supplementary Information 5, we clarify the relationship of our Eqs. (3) and (4) to the well-known expression for the likelihood of an inhomogeneous Poisson process. In Supplementary Information 6, we derive the modified Fokker-Planck equation Eq. (6) through the Feynman-Kac formula.

Eq. (3) (which is Eq. (4) in the revised manuscript) is a factorization of the joint probability density of data and a discretized trajectory $X(t)$. As the reviewer points out, the probability of a spike to occur at any specific time t_i is 0, but the probability density, defined as the ratio of the probability to spike in an interval Δt over Δt in the limit $\Delta t \rightarrow 0$, is finite and equal to the instantaneous firing rate $\lambda(t_i) = f(x(t_i))$ (see Chapter 1, Appendix C in Dayan & Abbott, 2001). If the continuous latent trajectory $X(t)$ is fixed and fully observed, then the factorization in Eq. (3) reduces to the likelihood for an inhomogeneous Poisson process with the known instantaneous firing rate $\lambda(t) = f(X(t))$ (see Supplementary Information 5). This formula is widely used for parameter estimation in neural encoding models, in which firing rate is modeled as a known parametric function of a fully observed stimulus (e.g., Paninski, 2004). In our case, however, the trajectory that produced the data is unknown and hence the Poisson likelihood formula cannot be applied directly as in Paninski, 2004. Instead, we need to consider all possible latent paths weighted according to how consistent they are with the data and with the Langevin dynamics.

To average over all possible latent paths, we derive the modified Fokker-Planck equation Eq. (6) using the Feynman-Kac formula, which converts the expectation of a stochastic process into a PDE. The reviewer is right that the Feynman-Kac formula relates the expected value of a certain stochastic process to the backward Kolmogorov equation. In Supplementary Information 6, we now show how this backward Kolmogorov equation can be transformed into the forward Fokker-Planck equation, which we use in all our subsequent calculations.

2) The conclusions in the 1st paragraph of page 5 ("To reveal how each non-stationary component contributes to the inference...") show a potentially serious challenge to the applicability of the model. Most importantly, I did not understand the authors' explanation for how the log likelihood of the fitted model is better than ground truth in Fig 3b. If this is really the case, it's unclear how or if this method could be used for model comparison/selection. As part of this, the distinction between the absorption operator and the absorbing boundary conditions needs to be explained more here ("the importance of the absorption operator by performing the inference with the initial distribution $p_0(x)$ and absorbing boundary conditions, but omitting the absorption operator").

Reply:

We clarified the explanation of results in Fig. 3b-d and the figure caption. In Fig. 3b-d, we perform the inference omitting the key non-stationary components to illustrate how each of them contributes to the accurate inference. For example, in Fig. 3b we perform the inference with the absorbing boundary conditions but omitting the absorption operator in the likelihood calculation Eq. (4). Such inference is

technically incorrect and would never be used in practice. The only purpose of this simulation is to demonstrate the importance of the absorption operator for accurate inference. In Fig. 3b, the likelihood is lower for the ground-truth potential than for the potential with the spurious barrier because the absorption operator was omitted in the likelihood calculation for both models with the ground-truth and fitted potential. This result shows that the absorption operator is necessary for accurate inference and does not challenge the model applicability.

We further explained the distinction between the absorbing boundary conditions and the absorption operator in the text related to Fig. 3b-d. The absorbing boundary conditions ensure that the likelihood only includes trajectories that do not reach the domain boundaries before the trial end time t_E . Such trajectories can terminate anywhere in the latent space at the trial end. The absorption operator, on the other hand, ensures that the likelihood only includes trajectories terminating at the domain boundaries at time t_E , irrespective of whether they also reached the boundary at an earlier time. When the absorbing boundary conditions and the absorption operator are used together, they ensure that the likelihood only includes trajectories reaching the boundaries for the first time at the trial end t_E , as it is required for systems in which reaching the boundaries terminates the observations. We illustrate the importance of the absorption operator and of the absorbing boundaries in Fig. 3b and 3c, respectively.

3) The limitations of the current method to address model comparison challenges appear overstated. On page 7, the authors state that “comparisons between simple parametric models proved ineffective to reveal the underlying neural dynamics.” This paper, however, demonstrates how modeling choices still exist even in more flexible approaches: does one include an absorbing bound? What is the dimensionality of the latent state? Are the dynamics constant across trials? Furthermore, the methods reported here do not report uncertainty in the model fit (many potentials explain the data similarly, even in simulation) or provide ways to quantitatively ask how well the dynamics in the data reflect different hypotheses (as comparing simple parametric models is designed to do). Thus, I think the authors need to be more exact and tone down how much this approach can do to solve the model comparison/selection problem.

Reply:

We expanded the discussion on page 8 to explain the advantages of our non-parametric framework over the conventional model comparison with a discrete set of alternative hypotheses. For the case of decision making, previous work compared parametric models implementing the “ramping” and “stepping” hypotheses for neural dynamics. However, an obvious pitfall is that neither of these two specific hypotheses may be correct. In contrast, our framework provides a flexible non-parametric description of dynamics, which covers a continuous space of hypotheses where the ramping and stepping dynamics are just special cases. Our framework can smoothly interpolate between many qualitatively different dynamics all expressed with the same analytical equations. This flexible approach enables discovering new hypotheses by fitting data, thus going beyond the classical model comparison, as we discuss in detail in our previous work (Genkin & Engel, 2020).

The reviewer is right that any model inevitably involves abstractions, assumptions, and modeling choices. This limitation is general, not specific to our framework. The modeling choices are informed by knowledge of the physical system and pragmatic considerations trading off details, tractability, and interpretability. For the example of decision-making dynamics, the experimental design dictates the choice of absorbing versus reflecting boundaries. In a reaction-time task, the animal terminates each trial, hence absorbing boundaries are used. In a fixed-duration task, the experimenter terminates each trial, hence reflecting boundaries are used. One-dimensional dynamics is a parsimonious choice for modeling two-alternative decision making, which is used by the overwhelming majority of previous models, including the prominent drift-diffusion as well as ramping and stepping models. Our choice of one-dimensional dynamics facilitates a comparison with these previous models, but can be extended to higher-dimensional dynamics in the future. Finally, the assumption that the dynamical model is constant across trials is reasonable when animal’s behavior is stable throughout the session. This assumption is also made by the majority of decision-making models and can be relaxed when necessary. All these considerations

are important to keep in mind when interpreting modeling results, but they do not undermine the advantages of our flexible framework over the conventional model comparisons.

The key advance of this paper is to develop the inference framework for non-stationary dynamics. We deliberately limit the discussion of model selection, interpretation, and uncertainty quantification, since all these important issues are covered in depth in our previous work (Genkin & Engel, 2020). In particular, Fig. 6e and Extended Data Fig. 6 in Genkin & Engel, 2020 shows a method for uncertainty quantification. In this paper, we only briefly discuss these issues and refer to our previous work for details, as otherwise, we feel, it would divert the focus from the main advances of our current work.

4) *The optimization process used to fit the model needs to be better defined. The plots show the log likelihood trajectory and optimization at a few epochs. It's not clear why these epochs were chosen – and a well-defined stopping criterion ought to be applied (not just that the term looked good at certain epochs). Additionally, it's not clear what the purpose of showing the optimization at several epochs is in the current paper. In the author's previous paper, they proposed an early stopping method for their model, and thus it made more sense to show the gradient decent process. Here, it's confusing to see several suboptimal models, and I would suggest removing them. Lastly, the authors state on page 5 that model selection is needed to identify “the correct model when the ground truth is not known”. No “correct” model exists in real data, and I would suggest rewording this to make it more clear how fitting should be approach when using real data.*

Reply:

We agree that in Fig. 4 it is not necessary to show fitted models at several iterations, and we removed these intermediate models from this figure. Now we only show the fitted models at the iteration when the likelihoods of the fitted and ground-truth models are equal, using the same consistent criterion in all panels. In Fig. 3, we decided to keep fitted models at several iterations for two reasons. First, Fig. 3a helps the reader to visualize how the potential shape smoothly changes through optimization. Our non-parametric approach is not common, and seeing how the potential morphs from the initial to final shape allows the reader to appreciate the flexibility of the non-parametric model. Second, Fig. 3b-d show the spectrum of potentials produced when the key non-stationary components are omitted in the inference. In this case, seeing how erroneous features develop through optimization helps to understand how each non-stationary component uniquely contributes to the accurate inference in Fig. 3a.

When referring to the situation with unknown ground truth, we changed the phrase “the correct model” to “the model that accurately approximates dynamics in the data”. We agree with the reviewer that modeling real data is a delicate issue. We carefully addressed this issue in our previous work (Genkin & Engel, 2020), where we introduced an operational definition of the correct model when the ground truth is not known. As mentioned in response to point 4, we only briefly mention model selection and interpretation in this paper and refer to our previous work for details, to keep the focus of this work on non-stationary dynamics.

5) *The use of the word “nonstationary” throughout the paper was, in my opinion, often a poor word choice. The new terms being estimated appear to be constant across trials: the draws from the initial state distribution are different across trials, but that distribution itself is the same. I think it would be more clear to label these as sources of trial-to-trial variability instead of non-stationarity to make it clear that the potential (or other terms) doesn't depend on time. (Non-stationary terms could be, for instance, gain changes in firing rate that occur slowly over trials).*

Reply:

Our use of the term “non-stationary dynamics” is standard in physics. For non-stationary dynamics, the probability density $p(x, t)$ depends on time within a trial. At the trial start, the probability density $p(x, t_0) = p_0(x)$ is different from the equilibrium density $p_{eq}(x)$, and it transiently evolves throughout the trial as illustrated in Fig. 2, third column. In contrast, for stationary dynamics, the probability density is time independent $p(x, t) = p_{eq}(x)$. We decided to keep this terminology since it is used in standard

textbooks, e.g., Risken 1995, Gardiner 1985. Thus, we consider non-stationary dynamics on a fast timescale within a trial, which is different from the trial-to-trial variability due to sampling from $p_0(x)$ at the trial start. A potential that changes across trials would introduce an additional source of non-stationarity on a slower timescale, but such slow non-stationarities are outside the scope of this paper. We clarified this issue in the Discussion section.

6) *The authors should address related methodology for estimating single-trial dynamics and possibly present a comparison to one or more other approaches (e.g., Nassar et al. "Learning Structured Neural Dynamics From Single Trial Population Recording." 2018 52nd Asilomar Conference on Signals, Systems, and Computers. IEEE, 2018.)*

Reply:

We now discuss this and other related methods in the Discussion section.

7) *The rigorousness of the approach could be improved by providing some limits on the convergence of the estimator (what can be estimated). The current paper suggests that any continuous potential can be fit, but without proof this feels too good to be true. What kinds of functions $\Phi(x)$ can be estimated given the uncertainty of how x can be observed through sparse spiking? I realize that strong proofs are beyond the scope of this paper, but I would like some guidelines for realistic inference - even really handwavy ones - to recognize this problem (e.g., if Φ isn't too wiggly, as was the case in your examples).*

Reply:

Indeed, the inference accuracy depends on the amount of available data and on the complexity of the underlying dynamics. In this paper, we consider cases where the inference accuracy is not limited by the data amount to isolate how it depends on non-stationary components. A large data amount generally results in more accurate inference (Fig. 4a,b). With insufficient data, the inference can underfit, i.e. not discover all features present in the system's dynamics. Inferring more complex dynamics requires larger data amounts. We clarified this issue in the Discussion section, referring for details to our previous work where we studied it systematically (Genkin & Engel, 2020). In particular, Supplementary Tables 2,3 in Genkin & Engel, 2020 show a comprehensive summary of results from 270 simulations with varying data amount, complexity of dynamics, and noise level.

Minor points

1) *Page 7, first paragraph: "Our framework ... can accommodate different observation processes, e.g., non-additive Poisson observation noise." Suggest rewording to say it accommodates Poisson observations as the integration approach here strongly depends on it being Poisson and no other noise models have been demonstrated.*

Reply:

We revised this sentence accordingly.

2) *Page 7, first paragraph: "Our framework generalizes to several latent dimensions". Given that this is the second paper the authors have written on this method, it would be good to see a 2D latent example to show that this claim is practical, not just theoretical.*

Reply:

We revised this sentence to state that our framework "can be generalized to several latent dimensions". While scaling our framework to 2D would be interesting and is conceptually straightforward using well-established 2D spectral elements method (Deville et al. 2002), it is beyond the scope of this work. In addition, focusing on one-dimensional dynamics facilitates a comparison with previous ramping and stepping models of decision making (see response to the point 3). Therefore, we defer 2D extension of our framework to future work.

References:

Bishop C. M. *Pattern Recognition and Machine Learning*. Springer (2006).

- Dayan, P. & Abbott, L.F. *Theoretical Neuroscience: Computational and Mathematical Modeling of Neural Systems*. The MIT Press (2001).
- Deville, M. O., Fischer, P. F., Fischer, P. F., & Mund, E. H. *High-order Methods for Incompressible Fluid Flow* (No. 9). Cambridge University Press (2002).
- Genkin, M. & Engel, T. A. Moving beyond generalization to accurate interpretation of flexible models. *Nat. Mach. Intell.* 2, 674–683 (2020).
- Haas, K. R., Yang, H. & Chu, J. W. Expectation-maximization of the potential of mean force and diffusion coefficient in Langevin dynamics from single molecule FRET data photon by photon. *J. Phys. Chem. B* 117, 15591–15605 (2013).
- Nassar, J., Linderman, S. W., Zhao, Y., Bugallo, M., & Park, I. M. Learning structured neural dynamics from single trial population recording. In *52nd Asilomar Conference on Signals, Systems, and Computers*, pp. 666-670. IEEE (2018).
- Paninski, L. Maximum likelihood estimation of cascade point-process neural encoding models. *Network: Computation in Neural Systems*, 15, 243-262 (2004).
- Risken, H. *The Fokker-Planck Equation*. Springer (1996).
- Gardiner, C. W. *Handbook of Stochastic Methods*. Springer (1985).
- Wei, C. & Shu, H. Maximum likelihood estimation for the drift parameter in diffusion processes. *Stochastics* 88, 699–710 (2016).
- Wei, C. Estimation for parameters in partially observed linear stochastic system. *IAENG International Journal of Applied Mathematics* 48, 123–127 (2018).

Reviewers' Comments:

Reviewer #1:

Remarks to the Author:

The authors have addressed the issues that were previously raised. I think the current form of the manuscript is acceptable.

Reviewer #2:

Remarks to the Author:

The authors have made several revisions and additions to their original paper. However, the revisions have not adequately addressed a substantial portion of my original concerns. As a result, I am unfortunately unable to recommend acceptance.

The detailed points of my assessment are below. The numbers correspond to points raised in the first review.

1. The additions to the supplement were helpful, but issues with the presentation of the likelihood in the methods in the main text remain. The authors' response to this concern repeated the same errors about the point process likelihood. The observation probability $p(y_t|x_t)$ is flatly not a probability, as the authors insist in the text. The transition likelihood $p(x_t|x_{t-1})$ as written does not include the fact that no spikes are emitted during that time period; conditioning on that information is required. The issue is again that this is a continuous time Poisson process. Factorisations that resemble this one are common – and perfectly correct – in binned time models, but that's not what's going on here.

The Poisson process probability does in fact show up in eq. 3, and I did not follow the argument in response given in the authors' rebuttal. The joint probability $P(X,Y)$ can be written $P(Y|X)P(X)$ where $P(Y|X)$ is the Poisson likelihood. The integral just marginalises over that unknown rate with the $P(X)$ part. This is the common Cox process formulation where a Cox process is a Poisson process with an unknown rate.

2. The results in Figure 3b are much clearer given the new explanation. But I'm still a bit perplexed about the utility of this comparison. The parameters of a model (here, the potential) can only really be interpreted within the context of the full model. One really shouldn't expect to recover the original potential after placing it into a different model (one without the absorbing boundary). More explanation could help illuminate how this exercise demonstrates the usefulness of the model.

3. While model comparisons are limited to the particular models chosen, this method appears limited to the specific point estimate recovered in one class of flexible model. There is nothing wrong with using flexible models – but the theme of this paper (and the authors' previous work) makes very strong insinuations about interpretability of this point estimate without any attempt to quantify model uncertainty. In the classic Bayesian parlance, this would be performing inference in the M-closed regime when the actual circumstances are M-complete (or even M-open). Thus, much of the emphasis on correct interpretation and discovery may instead unintentionally promote overinterpretation, and more care should be given in presentations of this approach to discourage poor statistical methodology with flexible models.

For example, it is stated in the discussion that the model overcomes "the brittleness of parametric model selection", but it's not really demonstrated that it overcomes these issues. Instead, flexible models are another tool, in addition to model comparison, to help better analyse and understand data.

4. This point was improved somewhat, but proof of the applicability of this model requires a method that doesn't have access to the ground-truth or oracle model in order to stop optimisation (it's ever so slightly circular to say that the inferred potential is similar when the GD trajectory intersects the level set of the likelihood at ground truth).

While the phrase "the model that accurately approximates dynamics in the data" is an improvement, a more accurate statement could be "the model that best fit the data".

The remaining points were addressed.

Reviewer #1 (Remarks to the Author):

The authors have addressed the issues that were previously raised. I think the current form of the manuscript is acceptable.

Reply:

We thank the reviewer for insightful comments that helped us to improve the manuscript. We are happy that the changes we made were satisfactory.

Reviewer #2 (Remarks to the Author):

The authors have made several revisions and additions to their original paper. However, the revisions have not adequately addressed a substantial portion of my original concerns. As a result, I am unfortunately unable to recommend acceptance. The detailed points of my assessment are below. The numbers correspond to points raised in the first review.

Reply:

We thank the reviewer for these further questions, which we addressed with substantial additional work. We believe these new revisions strengthened the results and helped improve the presentation clarity.

1) The additions to the supplement were helpful, but issues with the presentation of the likelihood in the methods in the main text remain. The authors' response to this concern repeated the same errors about the point process likelihood. The observation probability $p(y_t|x_t)$ is flatly not a probability, as the authors insist in the text. The transition likelihood $p(x_t|x_{t-1})$ as written does not include the fact that no spikes are emitted during that time period; conditioning on that information is required. The issue is again that this is a continuous time Poisson process. Factorisations that resemble this one are common – and perfectly correct – in binned time models, but that's not what's going on here.

The Poisson process probability does in fact show up in eq. 3, and I did not follow the argument in response given in the authors' rebuttal. The joint probability $P(X,Y)$ can be written $P(Y|X)P(X)$ where $P(Y|X)$ is the Poisson likelihood. The integral just marginalises over that unknown rate with the $P(X)$ part. This is the common Cox process formulation where a Cox process is a Poisson process with an unknown rate.

Reply:

We meticulously went through the main text and supplementary information to make sure that everywhere we refer to $p(y|x)$ and $p(x_{t+1}|x_t)$ as probability density. The transition probability density $p(x_{t+1}|x_t)$ incorporates the fact that no spikes are emitted during each interspike interval, which we further stressed in the revised Methods. Specifically, $p(x_{t+1}|x_t)$ is a solution of the modified Fokker-Planck equation (Eq. 6), which contains a term that accounts for the probability decay due to spike emissions. We derive Eq. 6 in Supplementary Information 6. Thus, the factorization in Eq. 4 is valid for a continuous time Poisson process and accounts for the absence of spikes during interspike intervals. Similar factorizations indeed exist for the likelihood of a binned Poisson process, with the difference that in our case the likelihood is a probability density of the observed spike sequence. In continuous time, the probability of any precise spike sequence $\{t_1, t_2, \dots, t_N\}$ is infinitesimal. We can obtain the probability of observing a spike within dt of each of the times $\{t_1, t_2, \dots, t_N\}$ by multiplying our likelihood by dt^N . We clarified this issue in Methods.

The reviewer is correct that likelihood calculation involves marginalization of the joint probability density $P(X, Y) = P(Y|X)P(X)$. The probability density $P(Y|X)$ is a Poisson likelihood of the observed spikes with firing rate that depends on a specific continuous trajectory \mathcal{X} . The form of our likelihood is similar to the likelihood of a Cox process (Supplementary Information 5). The main challenge of computing the likelihood is the marginalization over latent states. We perform the marginalization over continuous latent trajectories \mathcal{X} using a modified Fokker-Planck equation Eq. (6) for each interspike interval and then by numerical integration over a discrete set of remaining states $x_{t_1}, x_{t_2}, \dots, x_{t_N}$.

2) The results in Figure 3b are much clearer given the new explanation. But I'm still a bit perplexed about the utility of this comparison. The parameters of a model (here, the potential) can only really be interpreted within

the context of the full model. One really shouldn't expect to recover the original potential after placing it into a different model (one without the absorbing boundary). More explanation could help illuminate how this exercise demonstrates the usefulness of the model.

Reply:

Our framework includes three non-stationary components: initial state distribution, boundary conditions, and absorption operator (Fig. 1b). Results in Fig. 3 show how each of these components uniquely contributes to accurate inference. The necessity of all components for accurate inference is not that obvious. Spike trains generated from stationary vs. non-stationary dynamics appear similar (Fig. 2). Thus, one could assume that omitting non-stationary components may affect the inference only insignificantly. The results in Fig. 3 show that this is not the case: the inference accuracy deteriorates dramatically when non-stationary components are missing. Moreover, Fig. 3 demonstrates the unique roles of non-stationary components, each of which filters out trajectories with specific termination times in the model likelihood. We explain these results in the main text.

3) While model comparisons are limited to the particular models chosen, this method appears limited to the specific point estimate recovered in one class of flexible model. There is nothing wrong with using flexible models – but the theme of this paper (and the authors' previous work) makes very strong insinuations about interpretability of this point estimate without any attempt to quantify model uncertainty. In the classic Bayesian parlance, this would be performing inference in the M-closed regime when the actual circumstances are M-complete (or even M-open). Thus, much of the emphasis on correct interpretation and discovery may instead unintentionally promote overinterpretation, and more care should be given in presentations of this approach to discourage poor statistical methodology with flexible models.

Reply:

We agree that quantification of the model uncertainty is important when analyzing real data, as in the M-complete or M-open setting any discovered model is necessarily an approximation of reality. We therefore performed additional simulations to demonstrate the uncertainty quantification of the inferred model via bootstrapping (new Fig. 5d,e). Our method of uncertainty quantification does not require knowledge of the ground-truth model and is therefore applicable to real data.

For example, it is stated in the discussion that the model overcomes “the brittleness of parametric model selection”, but it's not really demonstrated that it overcomes these issues. Instead, flexible models are another tool, in addition to model comparison, to help better analyse and understand data.

Reply:

We revised the sentence accordingly: “Our framework can smoothly interpolate between many qualitatively different dynamics, which are all expressed with the same analytical equations, offering a powerful alternative to parametric model selection”.

4) This point was improved somewhat, but proof of the applicability of this model requires a method that doesn't have access to the ground-truth or oracle model in order to stop optimisation (it's ever so slightly circular to say that the inferred potential is similar when the GD trajectory intersects the level set of the likelihood at ground truth).

Reply:

Prompted by the reviewer's feedback, we included new results which demonstrate a model selection method that does not require knowledge of the ground truth (new Fig. 5 and Supplementary Fig. 2). We developed the model selection method based on feature consistency in our previous work (Ref. [2]), and we now extended this method to non-stationary dynamics. We derived an analytical expression for the feature complexity for non-stationary models and developed a method to quantify feature consistency for non-stationary dynamics (new Supplementary Information 7 and revised Methods). We show that model selection based on feature consistency can accurately identify the correct model for different ground-truth dynamics (new Fig. 5a-c and Supplementary Fig. 2). Moreover, our model selection method can be combined with bootstrapping to quantify the uncertainty of the inferred model (new Fig. 5d,e).

While the phrase “the model that accurately approximates dynamics in the data” is an improvement, a more accurate statement could be “the model that best fit the data”.

Reply:

“The model that best fits the data” usually means the model with the best likelihood (or validated likelihood). Our previous work (Ref. [2]) shows that such models often exhibit spurious features, i.e. do not accurately capture dynamics in the data. Therefore, we kept our original sentence unchanged.

Reviewers' Comments:

Reviewer #2:

Remarks to the Author:

Most of my concerns have now been addressed, and I believe it's suitable for publication without any additional review.

In particular, the additional material in figure 5 will help guide practitioners interested in this method. However, I have one remaining comment – my previous comment 1 - that I strongly insist the authors consider before final publication.

I am perplexed about the authors' insistence for including the factorisation in equation 4, in particular calling $p(y_t|x_t)$ a probability density. Writing out the pieces that you use to solve the marginalisation in such a factorised form is great, but $p(y_t|x_t)$ is still neither a probability distribution nor a density! Also, $p(x_t|x_{t-1})$ depends both on no spikes occurring between $t-1$ and t and spikes at $t-1$ and t . I believe this factorisation comes from the Hass et al paper, but I don't think it's correct there either: those authors really wanted to refer to Bayesian graphical models for some reason. Mathematical correctness here, not calling these pieces probabilities, wouldn't affect the results or methods and it requires very little effort so I cannot understand the stubborn refusal to correct this after two revisions. Moreover, a reader could easily mistake the form of this factorisation to mean the model is a discrete time model (I did the first time through) because the factorisation is correct for a discrete time model: the authors should want to flaunt the fact that this is a continuous time setup.

Reviewer #2 (Remarks to the Author):

Most of my concerns have now been addressed, and I believe it's suitable for publication without any additional review. In particular, the additional material in figure 5 will help guide practitioners interested in this method.

Reply:

We thank the reviewer for many thoughtful comments which helped us improve the paper significantly. We are glad that the revisions we made, in particular new results in Figure 5, were satisfactory.

However, I have one remaining comment – my previous comment 1 - that I strongly insist the authors consider before final publication. I am perplexed about the authors' insistence for including the factorisation in equation 4, in particular calling $p(y_t|x_t)$ a probability density. Writing out the pieces that you use to solve the marginalisation in such a factorised form is great, but $p(y_t|x_t)$ is still neither a probability distribution nor a density!

Reply:

We agree that $p(y_t|x_t)$ is technically neither probability nor probability density, but it is a probability per unit time, i.e. rate. Indeed, $p(y_t|x_t)dt$ is the probability of observing a spike within a small dt of time t given the latent state x_t . We clarified this technical aspect in the revised Methods after Eq. (4).

Also, $p(x_t|x_{t-1})$ depends both on no spikes occurring between $t-1$ and t and spikes at $t-1$ and t .

Reply:

This is not correct. The transition probability density for the latent state does not depend on spikes at $t - 1$ and t , which is shown in the graphical model in Fig. 1b.

I believe this factorisation comes from the Hass et al paper, but I don't think it's correct there either: those authors really wanted to refer to Bayesian graphical models for some reason. Mathematical correctness here, not calling these pieces probabilities, wouldn't affect the results or methods and it requires very little effort so I cannot understand the stubborn refusal to correct this after two revisions. Moreover, a reader could easily mistake the form of this factorisation to mean the model is a discrete time model (I did the first time through) because the factorisation is correct for a discrete time model: the authors should want to flaunt the fact that this is a continuous time setup.

Reply:

The factorization in Eq. (4) is correct, but there is a subtlety. The likelihood of a spike sequence $\{t_1, t_2, \dots, t_N\}$ in continuous time is not probability nor a probability density, as the reviewer points out. We obtain the probability of observing a spike within dt of each $\{t_1, t_2, \dots, t_N\}$ by multiplying the likelihood with dt^N (see Methods). Thus, the likelihood is technically a probability per unit time to the power N . The factorization in Eq. (4) is then technically a probability density (over $N + 2$ latent states) per unit time to the power N . We obtain the probability of observing a spike within dt of each $\{t_1, t_2, \dots, t_N\}$ given that the latent state was within dx of each $\{x_0, x_1, x_2, \dots, x_N, x_{t_E}\}$ by multiplying the factorization in Eq. (4) by $dt^N dx^{N+2}$.

Although the reviewer is correct that the likelihood and the factorization in Eq. (4) are neither probability nor probability density, calling these objects “probability / probability density per unit time to the power N ” would be confusing rather than add clarity. Therefore, we take a pragmatic decision to explain what these objects are and then refer to them as probability density. Note, that this issue is general for point processes and different authors refer differently to the object $p(t_1, t_2, \dots, t_N)$ similar to our likelihood. For example, Dayan & Abbot call $p(t_1, t_2, \dots, t_N)$ for the inhomogeneous Poisson process a “probability density” (page 23 in [1]), whereas Stratonovich calls $p(t_1, t_2, \dots, t_N)$ for a general point process a “distribution function” (page 144 in [2]).

[1] Dayan & Abbott. *Theoretical Neuroscience: Computational and Mathematical Modeling of Neural Systems*. The MIT Press, 2005.

[2] Stratonovich. *Topics in the Theory of Random Noise*. Gordon and Breach, New York, 1963.